# Torso Shape Improves the Prediction of Body Fat Magnitude and Distribution

**DOI:** 10.3390/ijerph19148302

**Published:** 2022-07-07

**Authors:** Simon Choppin, Alice Bullas, Michael Thelwell

**Affiliations:** Advanced Wellbeing Research Centre, Sheffield Hallam University, 2 Old Hall Road, Sheffield S9 3TU, UK; a.bullas@shu.ac.uk (A.B.); m.thelwell@shu.ac.uk (M.T.)

**Keywords:** metabolic syndrome, anthropometry, 3D body scan, body shape, fat volume, fat distribution, multiple linear regression

## Abstract

Background: As obesity increases throughout the developed world, concern for the health of the population rises. Obesity increases the risk of metabolic syndrome, a cluster of conditions associated with type-2 diabetes. Correctly identifying individuals at risk from metabolic syndrome is vital to ensure interventions and treatments can be prescribed as soon as possible. Traditional anthropometrics have some success in this, particularly waist circumference. However, body size is limited when trying to account for a diverse range of ages, body types and ethnicities. We have assessed whether measures of torso shape (from 3D body scans) can improve the performance of models predicting the magnitude and distribution of body fat. Methods: From 93 male participants (age 43.1 ± 7.4) we captured anthropometrics and torso shape using a 3D scanner, body fat volume using an air displacement plethysmography device (BODPOD^®^) and body fat distribution using bioelectric impedance analysis. Results: Predictive models containing torso shape had an increased adjusted R^2^ and lower mean square error when predicting body fat magnitude and distribution. Conclusions: Torso shape improves the performance of anthropometric predictive models, an important component of identifying metabolic syndrome risk. Future work must focus on fast, low-cost methods of capturing the shape of the body.

## 1. Introduction

The prevalence of obesity continues to increase globally; rates have doubled in over 70 countries since 1980. As of 2018, at least 30% of men and 35% of women were classified as obese in many countries worldwide [1]. Obese individuals are at a greater risk of developing metabolic syndrome, a cluster of conditions including dyslipidemia, insulin resistance and hypertension, and are associated with an increased risk of type-2 diabetes mellitus (T2DM) and cardiovascular disease (CVD) [2]. Correctly identifying individuals with metabolic syndrome, or those at risk of developing its symptoms, is vital to ensure interventions and treatments can be prescribed as early as possible, improving clinical outcomes, reducing long term costs and maximising quality of life.

In practice, obesity is defined using the body mass index (BMI) and stratified into categories according to the World Health Organization (WHO) [3]. Although there are many benefits to using BMI, particularly for population-based screening, evidence indicates the existence of obesity subgroups such as the metabolically healthy but obese, or the metabolically unhealthy but normal weight [4]. Obesity is a heterogeneous condition; susceptibility to metabolic complications is a function of both the magnitude and distribution of an individual’s bodily fat mass—particularly among those of normal body weight [1,5,6]. Consequently, people with the same BMI can have varied cardiovascular and metabolic risk profiles, making true health risk difficult to determine [1,7]. Previous studies have suggested that obesity-related health risks are disproportionately due to accumulations of abdominal fat—both visceral and deep subcutaneous—rather than peripheral subcutaneous fat located in the limbs [8]. Central obesity is defined as an excess accumulation of fat in the abdominal area, particularly visceral fat [9].

Currently, practitioners conducting clinical and population-based health screenings rely upon traditional anthropometrics (and derived indices) to estimate BF% and visceral adipose tissue, and to classify individuals according to cardio-metabolic risk [1,9]. It has been shown that combining BMI with additional proxies of central obesity, such as waist girth and waist–hip ratio (WHR) can refine obesity classification when assessing cardio-metabolic risk [9,10,11,12]. As an anthropometric parameter, waist girth may be more effective than BMI at identifying individuals at risk from metabolic syndrome, even in lean populations [13], but it still has limitations. Differences in bodily fat distribution between ethnicities makes allocating specific cut-off values complicated and studies have highlighted the presence of non-Caucasian populations with ’normal’ waist circumferences but indicators of the metabolic syndrome [14,15]. Current approaches to anthropometry could be improved by adopting more sophisticated measures that improve population-level obesity assessment and health risk categorisation [16].

Three-dimensional (3D) imaging systems capture detailed 3D point cloud data of the body’s surface, from which size and shape characteristics can be extracted. Measures obtained from 3D imaging have been used to describe, interpret and analyse the human body for applications in health [17,18]. However typical measures taken from 3D scans have been combinations of lengths, girths and commonly used health indices, such as the waist to hip ratio (WHR) [19,20], rather than ‘new’, advanced anthropometrics. Recent studies have used the detailed information contained in 3D scan images to derive alternative anthropometrics. This has included using methods such as principal components analysis (PCA) [21,22], surface curvature [23] and geometric morphometrics [24]. Thelwell et al., focusing on the torso, showed that ‘shape measures’ (obtained through geometric morphometrics) can describe variations between individuals that cannot be explained by standard anthropometrics [25]. Though it is currently unknown what these unexplained variations in torso shape represent in terms of physical health, it is posited that this additional information might identify subtle surface morphological features that are related to body fat distributions and associated health risks. If so, ‘shape measures’ should be able to complement existing anthropometrics with regards to predicting the magnitude and distribution of body fat—A key indicator of the metabolic syndrome.

The purpose of this study was to determine whether torso shape measures can improve the performance of predictive models with regards to the magnitude and distribution of body fat.

## 2. Materials and Methods

### 2.1. Participants

We recruited a convenience sample of 93 male participants (age 43.1 ± 7.4 years) who were stratified for BMI: normal weight (18.5 ≤ BMI ≤ 24.9 kg/m^2^), overweight (25.0 ≤ BMI ≤ 29.9 kg/m^2^) and obese (BMI ≥ 30 kg/m^2^) [10]. Participants were also controlled for age, so most participants (n = 91) were in the age range 30–60, an at-risk group with regards to obesity-related metabolic health issues. Participants were not screened for any other comorbidities.

Participants were instructed to avoid taking part in strenuous exercise 12 h before testing, avoid excessive food or fluid intake the day before testing and to not eat or drink or apply lotions or skin creams at least three hours before testing to ensure accuracy of results. Participants wore non compressive, form-fitting shorts for all measurement procedures, as well as a tight-fitting swimming cap for plethysmography measurement.

### 2.2. Data Acquisition

The data analysed in this study consisted of 3D body imaging scan data, bioelectrical impedance analysis (BIA) data and air displacement plethysmography (ADP). From each 3D scan we extracted:Anthropometrics of the torso:○Waist girth (minimum girth in torso),○Stomach girth (maximum girth between waist and hips),○Hip girth (greatest posterior protuberance of the buttocks),○Chest girth (maximum girth of upper torso/bust),○Torso length (vertical distance between xiphoid process and hip).Maximum girths of left and right thighs,Maximum girths of left and right biceps,3D shape parameters of the torso,Volumes of the torso and limbs,Surface areas of the torso and limbs,The anthropometric index WHT.5R ($waist\;girth/height^{0.5})$.

BIA measurements were used to calculate the trunk:peripheral fat ratio. ADP measurements were used to calculate the body fat percentage.

#### 2.2.1. 3D Imaging

The 3D image data of each participant was acquired using a Size Stream SS14 whole-body scanner (Size Stream, Cary, NC, USA), in accordance with ISO 20685-1:2018 [26].

During scanning, participants stood at the centre of the calibrated volume with their feet in a specified position. They held adjustable postural aids at the side of the scanning device (Figure 1a) adopting a ‘downward V’ scanning posture. To minimise postural sway, participants were asked to visually focus on a point at the front of the scanning area and breathe lightly during the short scan duration (~10 s). The scanner collected three consecutive scans—the 3D point cloud used in the study represented the median of the three scan images in terms of extracted measurements (Figure 1b).

Anthropometrics, surface areas and volumes were automatically extracted from each 3D image using proprietary Size Stream Body Scanner software (Size Stream, Cary, NC, USA). The raw body scan data were also used to extract torso shape parameters (described in Section 2.3.2).

#### 2.2.2. Whole-Body Air Displacement Plethysmography (ADP)

Whole-body air displacement plethysmography (ADP) was performed using the BODPOD^®^ Body Composition System (Life Measurement Instruments, Concord, CA, USA).

The BODPOD was calibrated prior to each collection. Room temperature was between 21–27 °C, not varying by more than ±0.5 °C during a test. Relative humidity was between 20–70%, not varying by more than ±5%. The door of the laboratory was kept closed during testing.

Participants sat inside the BODPOD, remaining still and breathing normally for at least two measurements (~50 s each). If the first two measurements differed by more than 150 mL, a third measurement was performed [27], the average of the closest two measurements was reported. Population specific equations were used to convert body volume to fat mass. Siri [28] was used for white and Asian participants, Schutte [29] was used for Black participants, while Brozek [30] was used for extremely lean or obese participants.

#### 2.2.3. Bioelectrical Impedance Analysis (BIA)

Bioelectrical impedance measures were collected using a Tanita MC-780 multifrequency segmental body composition analyser (MC-780MA, Tanita Corp., Tokyo, Japan). Participants stood in a stable, barefoot position on the analyser’s platform and held both handles with straight arms held slightly away from their body. Estimates of segmental fat mass and fat free mass values were given for the trunk, left and right arms and legs, as well as estimates of total body fat. Two repeat measures were taken for each participant, with the average used in subsequent analysis.

### 2.3. Data Analysis

#### 2.3.1. Body Measures

Traditional anthropometrics (lengths and girths) were normalized to account for the differing height of participants. Allometric scaling was assumed with the relationship lny=α+βlnH, where y is the measure, H is the participant’s height and α, β are parameters specific to each measure. Parameters were taken from [31] for measures of the arms and legs and [32] for measures of the torso.

Body volumes and surface areas were normalised by the height of each participant (allometric scaling coefficients were not available).

#### 2.3.2. 3D Imaging Data Post-Processing

A torso segment was defined as the area between the xiphoid process and the buttock landmark [33] (corresponding to the gluteal (hip) girth location). Using the xiphoid process as the torso’s superior boundary helped scan segmentation by avoiding occlusions in the axilla (armpit) region [24,34]. The vertical height of the xiphoid process was found to be 60% ± 1.5% of the distance between the buttock and neck height landmarks—which were automatically obtained from each 3D scan [25].

Non-shape variation (scale, location and orientation) was removed from the 3D scan data. First, an anatomical coordinate axis system was created at the centre of each torso segment, according to the convention defined in ISO 20685-1:2018 [26]. The x-axis was aligned in the sagittal direction, the y-axis was aligned in the transverse direction and the z-axis was aligned longitudinally (Figure 2). The centre-line of the torso passed through the midpoint of the xiphoid and spinous processes and the origin was located at the bottom of the torso.

Following alignment, points outside of the torso region were removed from each 3D image. Twenty-five separate, 2 mm bands of 3D data points were extracted from each torso at uniform intervals along its length [35]—creating a series of 2D-shape profiles along the length of the torso. Finally, these shape profiles were rescaled according to centroid size to remove the effects of scale between participants in the sample [36].

#### 2.3.3. Torso Shape Feature Extraction

Shape features were extracted from each scaled torso using cubic smoothing splines, discrete fast-Fourier transforms (FFT) and principal components analysis (PCA). Initially, splines were fit to shape profiles—reducing the complexity of each profile while preserving shape. Second, frequency content was extracted from each profile using individual FFTs—preserving only the lowest 10 complex coefficients from each (resulting in 250 coefficients for each torso segment). Dimensionality was further reduced by transforming the entire set of FFT coefficients. Principal components of torso shape were derived from a large dataset of ~5000 males in previous work [25], eigenvectors for the first 10 principal components were used to obtain the shape parameters of each torso. A detailed description of this process is available in the paper by Thelwell et al. [24]. This resulted in 10 independent shape parameters describing each participant’s torso shape, PC1 to PC10.

### 2.4. Statistical Analysis

Histograms and Q-Q plots were visually inspected, and a Shapiro–Wilks test was conducted to assess the normality of body composition measures and extracted size and shape measures.

Pearson’s product-moment correlations, r, were used to assess collinearity between all measures of size and shape. This information was used to select independent parameters for each predictive model.

#### 2.4.1. Selection of Independent Parameters

Collinearity in model independent parameters was avoided as much as possible. As principal components, the torso shape parameters had very low amounts of collinearity—it was non-zero because the eigenvectors of the shape parameters were derived from a different dataset [25].

Collinearity existed in the anthropometrics, volumes and surface area parameters, this was minimised by creating new, derived measures and omitting parameters with high collinearity.

Of the size parameters, hip girth, torso length and WHT5R were included as a subset. Two new parameters were derived: average thigh girth and average bicep girth.

Of the volume parameters, torso volume was included as a subset and four new parameters were derived: average leg volume, average arm volume, torso:limbs volume ratio and legs:arms volume ratio.

Of the surface area parameters, torso surface area was included as a subset and four new parameters were derived: average leg surface area, average arm surface area, torso:limbs surface area ratio and legs:arms surface area ratio.

A summary of independent parameters and their correlation coefficients is given in Figure 3.

#### 2.4.2. Linear Regression Models

Six separate multiple linear regression models were formed to assess the predictive performance of shape, size (anthropometrics, volumes and surface areas) and combinations of size and shape. Each model contained 10 independent parameters. As each subset of size, volume and surface area parameters contained five distinct variables an ‘ideal’ subset of five shape parameters was chosen for each ‘shape and size’ model. To do this, an exhaustive search of each possible subset of 5 from 10 (252 unique subsets) was conducted. The subset responsible for the lowest mean squared error was chosen to be used in the main model (following the same stepwise procedure as described below).

The independent parameters of each model type were arranged as follows:Shape only: 10 shape parameters of the torso;Anthropometrics and volumes: 5 measures of the torso and limbs, 5 volumes of the torso and limbs;Anthropometrics and surface areas: 5 measures of the torso and limbs, 5 surface areas of the torso and limbs;Anthropometrics and shape: 5 measures of the torso and limbs, 5 ‘best performing’ shape parameters of the torso;Volumes and shape: 5 volumes of the torso and limbs, 5 ‘best performing’ shape parameters of the torso;Surface areas and shape: 5 surface areas of the torso and limbs, 5 ‘best performing’ shape parameters of the torso.

Two dependent parameters were modelled:

1. The proportion of fat mass in the body (as measured by ADP).

2. The trunk to peripheral fat ratio (TPFR): the ratio of fat volume in the torso to fat volumes in the peripheral (limbs in this case)—calculated using values from the BIA.

#### 2.4.3. Stepwise Regression Procedure

Stepwise regression was used (as in previous studies [21,24]) to allow each model to ‘compete’; stepwise regression provided a convenient means of creating sophisticated but transparent regression models maximising possible interactions between parameters.

Initially, linear models were created; the Akaike information criterion (AIC) was used to determine whether additional interaction terms should be added (forward stepwise regression) or existing terms should be removed (backward stepwise regression). No higher order terms or interactions were included. Terms were added to the model if they increased the AIC by any amount; terms were removed from the model if doing so increased the AIC by at least 0.01.

#### 2.4.4. Model Performance

Performance of the model was assessed by its adjusted R^2^ value and root mean squared error. To assess the generality of the model and examine over-fitting, a 10-fold cross-validation was performed, calculating root mean squared error (RMSE). The contribution of each term in the model was assessed by calculating their relative weights [37], which were examined in combination with *p*-values.

## 3. Results

Measured body fat percentage had a mean value of 21.02% with a standard deviation of 7.82% (range 6.55–42.86%).

Measured trunk:peripheral fat ratios had a mean value of 1.57 with a standard deviation of 0.2236 (range 0.84–2.09).

The characteristics of participants in the present analysis are shown in Table 1.

### 3.1. Subsets of Shape Parameters

Each predictive model that used shape parameters in combination with another metric used a subset of 5 shape parameters from the 10 available. The subset used for each model is shown in Table 2.

### 3.2. Performance of Predictive Models

Regarding the prediction of fat proportion, models containing both size metrics and shape parameters had a mean R^2^ value of 0.8347 compared to 0.7378 for models that did not contain shape parameters and 0.8331 for shape only. Cross-validation suggested the ability of a model to generalise. The worst drop in performance was observed by the ‘shape only’ model, where RMSE increased by 18.8% in cross-validation. By contrast, the best performing model, ‘size and shape’, only suffered a 6.1% increase in RMSE in cross-validation. A summary of results for all models is shown in Table 3.

Regarding the prediction of distribution of fat, models containing both size metrics and shape parameters had a mean R^2^ of 0.7105 compared to 0.5497 for models that did not contain shape parameters and 0.4904 for shape only. The ‘size and shape’ model saw the biggest drop of performance in cross-validation with an 18.45% increase in RMSE; however, cross-validated RMSE was lower than the trained RMSE for all other models. The ‘size and volume’ model saw the lowest increase in cross-validated RMSE at 6.8%. A summary of results for all models is shown in Table 4.

### 3.3. The Relative Weights of Terms in Each Model

The relative weights indicate the contribution of each term to a model’s performance. Table 5 and Table 6 show the three largest contributions to each model’s performance when predicting fat amount and fat distribution respectively. Nonsignificant terms have not been included.

## 4. Discussion

When predicting bodily fat proportion and distribution, shape parameters improve model performance. With fat proportion, all models containing shape outperformed models that did not—even the ‘shape only’ model that contained no measures of size. With fat distribution, increased performance was only observed when measures of shape and size were combined, the ‘shape only’ model exhibited the worst predictive performance. Best performance was obtained when combining measures of size and shape. Anthropometrics complemented shape best, with the highest performance (lowest RMSE values) observed in the ‘anthropometrics and shape’ models.

While performance was improved by combining size with shape for both model types (proportion and distribution), the greatest improvement was observed for fat distribution models, suggesting that measures of size are limited in their ability to predict fat distribution and that the additional information available in shape parameters is particularly suited to predict distribution of fat around the body.

However, the ability of models to predict fat distribution was generally lower than their ability to predict proportion of fat. The highest R^2^ in predicting proportion of fat was 0.848 (for the ‘shape and volume’ model) compared to 0.786 for distribution of fat (for the ‘shape and measures’ model). In addition, their ability to generalise was also lower and cross-validated RMSE values were proportionally higher. When predicting proportion of fat, models that combined shape and size had a cross-validated RMSE 7.5% higher than trained RMSE (on average). When predicting distribution of fat, their cross-validated RMSE were 14% higher than trained RMSE (on average). Previous studies that have explored the use of body shape parameters to predict fat proportion reported RMSE values similar to those in this study [21,23], although different technology (DXA) was used to obtain comparator fat proportion values.

This study competitively tested different model structures to see which could yield the highest predictive performance. Multiple linear regression models were chosen for this task not because they are most likely to result in the greatest performance, but because they are simple to interpret and still applicable to modelling in health domains. It is possible (and likely) that greater model performance could be achieved by using more sophisticated techniques such as artificial neural networks. It was important for this work that each of the models could be compared and started from the same basis—a linear combination of 10 independent parameters. It will be the work of future studies, when sample sizes are appreciably larger, to fully assess the predictive capability of shape parameters. With this small study, it should be acknowledged that we have focused on males only and were unable to obtain a representative distribution of ethnicities within the sample. This is particularly important because of known differences in the metabolic syndrome within different ethnic groups and how this relates to body fat amount and distribution.

Two shape parameters were shown to be key in the prediction of fat proportion and fat distribution. The PC2 shape parameter, which is best associated with fat proportion, tracks large-scale shape change; the PC7 shape parameter, which is best associated with distribution of fat, tracks much smaller-scale shape change. The effect of change in both parameters can be visualised in the recent paper by Thelwell et al. [25]. The paper also found that while a proportion of changes in shape can be explained by changes in a person’s size (around 50%), the remaining variation cannot. The ‘additional’ information contained within shape parameters could be the contributory factor which leads to the improved performance of these models. While it is not clear why PC2 and PC7 contributed the most, it is valuable to note that larger-scale shape changes contribute to the prediction of fat proportion while smaller-scale shape changes contribute to the prediction of fat distribution.

Given the established relationship between fat distribution and metabolic health risk [1], shape parameters could play a role in population level health assessment [17]. It would be challenging to implement 3D shape assessment in common practice because of the complexity associated with its measurement. This study has revealed that certain parameters have a more significant role in prediction than others (PC7 in the prediction of fat distribution, for example). An important future step will be to find simple, low-cost and quick ways of measuring 3D shapes.

Currently, shape parameters are derived from a 3D geometry of the torso; the process requires a 3D scan of the segment and subsequent algorithmic steps to define the final list of parameters. Future research should focus on how these parameters can be measured using simpler processes or tools. If measures taken from images (on a mobile phone, for example [38]) or given by simple manual tools can serve as proxies for shape parameters, they can feasibly be used as part of standard anthropometric assessments in clinical settings.

### Limitations

Errors seen in the predictive models will have been increased because of limitations with the equipment used. Both the BODPOD and BIA device use predictive equations when calculating proportion and distribution of fat. The reduced predictive ability of the fat distribution models could be due to the increased difficulty associated with predicting fat distribution; it could also be due to the known limitations of the BIA device used to measure the metric. The allometric normalization of anthropometrics will always be limited in that the scaling exponents used will usually have been derived from a different population. Future work should seek to assess predictive performance using alternative medical imaging devices such as magnetic resonance imaging and dual-energy X-ray absorptiometry where proportion and distribution of fat can be measured directly.

This study has been limited by its use of male participants only with a limited ethnic profile. Future work must include increased diversity and assess the suitability of this approach in a female population—an essential step for applicability to the general population.

## 5. Conclusions

Body shape improves the performance of anthropometric regression models.When predicting proportions of fat, larger-scale shape parameters are important for prediction (denoted as a lower-order parameter PC2, for example).When predicting distribution of fat, smaller-scale shape parameters are important for prediction (denoted as a higher-order parameter PC7, for example).Fat distribution is difficult to predict using size or shape alone but combining them increases predictive performance.Future research should investigate correlations between shape parameters and metabolic health risk.

## Figures and Tables

**Figure 1 ijerph-19-08302-f001:**
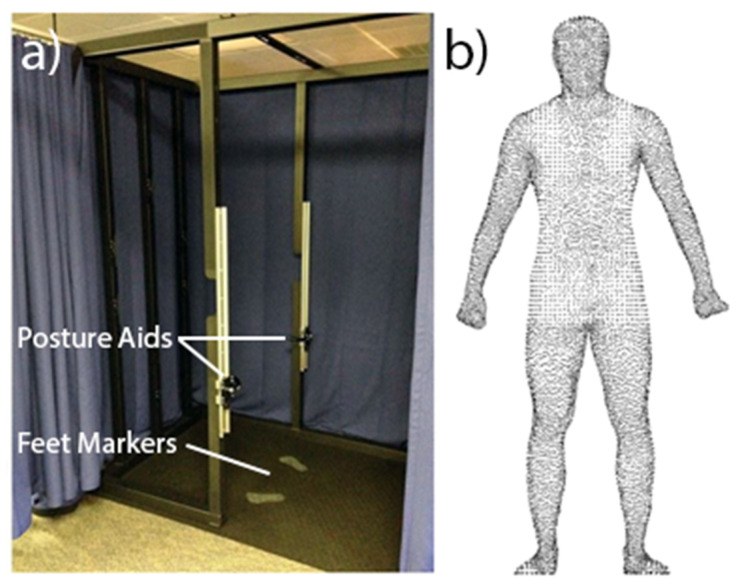
(**a**) Handheld posture aids within the scanner, (**b**) final scans were the median of three with respect to extracted anthropometrics.

**Figure 2 ijerph-19-08302-f002:**
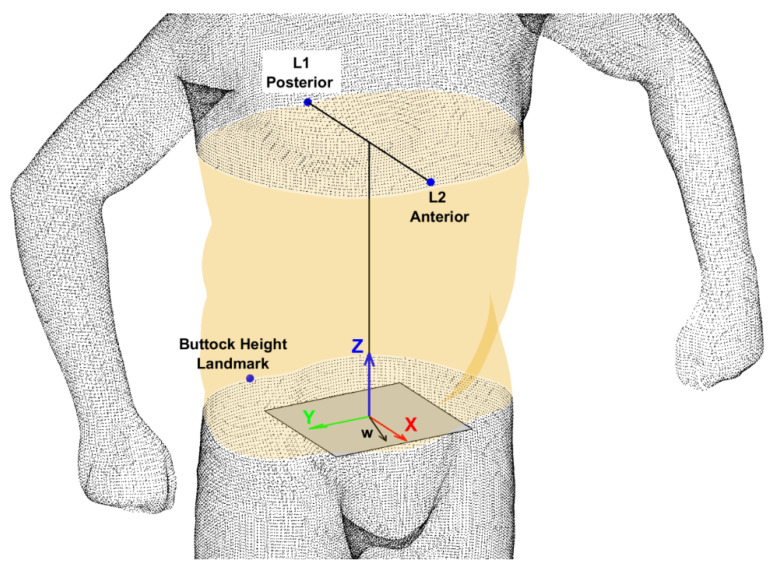
Landmarks used to segment the torso and create a local co-ordinate system within each 3D image, originally used in Thelwell et al. [25]. Please note, the scan data in this figure are for illustrative purposes only.

**Figure 3 ijerph-19-08302-f003:**
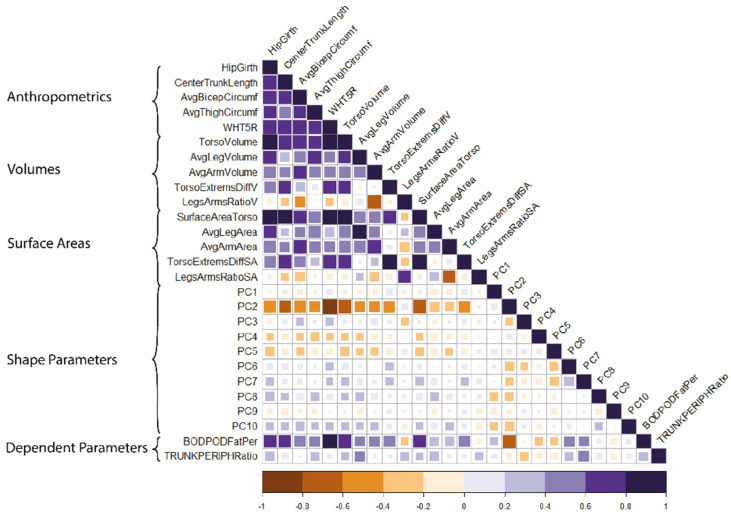
Higher correlations were observed between measures, volumes and surface areas than shape parameters, which were orthogonal by design.

**Table 1 ijerph-19-08302-t001:** Physical and demographic characteristics of participants.

	Mean (SD)	Min.	Max.
Number of participants, n = 93			
Physical characteristics			
Age (years)	43.1 (7.4)	28	58
Height (cm)	177.9 (6.8)	160.8	196.9
Weight (kg)	82.9 (13.3)	57.4	124.1
BMI (kg/m^2^)	26.2 (3.9)	20.3	38.3
WHR (cm/cm)	0.87 (0.06)	0.76	1.01
Total body fat (%)	21.02 (7.82)	6.55	42.86
Trunk:peripheral fat ratio (kg/kg)	1.57 (0.22)	0.84	2.09
Demographics			
Ethnicity: n (%)			
White	80 (86.0%)		
Asian or Asian British	6 (6.5%)		
Black, African, Black British or Caribbean	1 (1.1%)		
Mixed or multiple ethnic groups	4 (4.3%)		

**Table 2 ijerph-19-08302-t002:** The shape parameters used in each linear regression model. Each number corresponds to a specific shape parameter. For example, the regression model predicting body fat proportion with anthropometric parameters used shape parameters PC1, PC2, PC4, PC6 and PC7.

Independent Parameters	Shape Parameters Used When Predicting:	
	Proportion of Body Fat	Distribution of Body Fat
Anthropometrics and Shape	1 2 4 6 7	1 4 5 6 7
Volume and Shape	1 2 3 4 5	3 4 5 7 8
Surface Area and Shape	1 2 3 5 7	3 4 5 7 10

**Table 3 ijerph-19-08302-t003:** Performance of models predicting the proportion of body fat in percentage points.

Model	Adj. R^2^	RMSE	Cross-Val RMSE
Shape only	0.8331	3.19	3.79
Anthropometrics and Surface Area	0.7365	4.01	4.35
Anthropometrics and Volume	0.7390	4.00	4.27
Anthropometrics and Shape	0.8463	2.93	3.11
Shape and S.A.	0.8096	3.41	3.63
Shape and Volume	0.8482	3.05	3.35

**Table 4 ijerph-19-08302-t004:** Performance of models predicting the distribution of body fat as a ratio.

Model	Adj. R^2^	RMSE	Cross-Val RMSE
Shape only	0.4904	0.160	0.181
Anthropometrics and S.A.	0.5321	0.153	0.167
Anthropometrics and Volume	0.5673	0.147	0.157
Anthropometrics and Shape	0.7864	0.103	0.122
Shape and S.A.	0.6776	0.127	0.144
Shape and Volume	0.6676	0.129	0.142

**Table 5 ijerph-19-08302-t005:** Relative weights and associated significances for model terms predicting proportion of fat.

Model	Model Term	Relative Weight	*p*
Shape Only	PC2	41.0	<<0.001
(26 terms)	PC7	8.6	0.006
	PC1	7.3	<<0.001
Anthropometrics and S.A.	WHT.5R	32.5	<<0.001
(17 terms)	Torso S.A.	12.7	0.003
	Torso:Limbs S.A.	9.17	0.004
Anthropometrics and Volume	WHT.5R	23.1	<<0.001
(16 terms)	Avg. Bicep Girth	5.6	<<0.001
	Avg. Thigh Girth	3.9	0.04
Anthropometrics and Shape	PC2	16.8	<<0.001
(21 terms)	Hip Girth	10.0	0.004
	PC6	8.8	<0.001
Shape and S.A.	PC2	32.5	<<0.001
(19 terms)	Torso S.A.	11.18	0.04
	PC7	9.41	<<0.001
Shape and Volume	PC2	22.5	<<0.001
(25 terms)	Torso:Limbs Volume	11.3	0.02
	Torso Volume	9.9	0.05

Showing the top three, significant (*p* < 0.05) terms with regards to relative weight. Nonsignificant terms are not included.

**Table 6 ijerph-19-08302-t006:** Relative weights and associated significances for model terms predicting distribution of fat.

Model	Model Term	Relative Weight	*p*
Shape Only	PC7	16.63	0.009
(24 terms)	PC3	14.81	<0.001
	PC3 PC8	9.65	0.005
Anthropometrics and S.A.	Hip Girth	12.0	0.027
(17 terms)	Avg. Bicep Girth	11.3	<<0.001
	Torso S.A.	10.0	0.007
Anthropometrics and Volume	Hip Girth	11.02	0.035
(16 terms)	Avg. Bicep Girth	10.82	<<0.001
	Torso Volume	10.20	< 0.001
Anthropometrics and Shape	PC7	11.6	0.004
(29 terms)	Hip Girth	10.7	<<0.001
	Avg. Thigh Girth	10.2	<<0.001
Shape and S.A.	PC7	15.3	<<0.001
(25 terms)	PC3	11.0	<<0.001
	PC4 Torso S.A.	10.2	<<0.001
Shape and Volume	PC7	15.7	<0.001
(23 terms)	PC3	10.9	<<0.001
	PC4 Torso Volume	10.2	<<0.001

Showing the top three, significant (*p* < 0.05) terms with regards to relative weight. Nonsignificant terms are not included.

## Data Availability

Publicly available datasets were analysed in this study. These data can be found here: https://shurda.shu.ac.uk/id/eprint/148, accessed on 3 July 2022.

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
