# Peer review of "Torso Shape Improves the Prediction of Body Fat Magnitude and Distribution"

_ijerph, 2022, doi:10.3390/ijerph19148302_

Round 1

Reviewer 1 Report

The authors have assessed whether measures of torso shape (from 3D body scans) can improve the performance of models predicting the magnitude and distribution of body fat. Paper is written well but following needs to be addressed before acceptance:

1.     Please provide latest epidemiological data on obesity in the introduction section.

2.     Author did not mentioned inclusion and exclusion criteria in detail. Participant included in the study had any other comorbidities or not? How many male or female participants?

3.     Which method is used to calculate the sample size? The details of ethics committee and informed consent process should be mentioned in the methodology section.

4.     Table 1 should go in result section and not methodology.

5.     Many metabolic diseases are associated with obesity. Did authors included those type of participants in the study? If yes, please provide separate analysis for those participants.

6.     Discussion should include a paragraph highlighting limitations of the study.

7.     Language needs revision.

Author Response

Thank you for your review. I have attached a document with responses to your suggestions and points.

Reviewer 2 Report

Dear authors, dear editor,

The manuscript describes a new method to evaluate and measure obesity - a common risk factor for metabolic syndrome. Obesity is often measured by using BMI, however, it is widely known that BMI is not the golden standard to measure obesity and BMI cutoffs are criticized.

The manuscript describes a new method to combine anthropometric measurements and shape parameters to increase the predictive power.

I recommend the publication of the manuscript after a minor revision.

There are some inconsistencies and typing errors that need to be corrected before publication.

-          The manuscript does not have a clear research question and hypothesis. You name an aim of the study but do not give a testable hypothesis. I suggest adding a hypothesis that can be tested using statistical tests (as the authors did in their paper). Did you expect their method to improve predictability or not?

-          Minor typing errors/missing dots: lines 369, 193, 248

-          Line 265: % is missing

-          Throughout the whole manuscript “Error!” is written where I assume table 2 and table 3 should be mentioned.

-          Did you do backward or forward stepwise linear regression? Please mention the used method in the text.

-          Line 256 and 257: When AIC increased you added AND removed variables. That can’t be correct, please check.

-          Table 2: What do the numbers mean? Please explain as it remains unclear to the reader.

-          Table 3: What does the abbreviation S.A. mean? Please explain.

-          Line 279: The abbreviation RSME is not explained.

-          Estimation of percentage of body fat and distribution: The estimates are based on equations published elsewhere. I assume this is the best way to do it, however, it is also a limitation and should at least be mentioned! The same applies to the process of allometric scaling. You used published parameters but how correct are these? I agree that this is the best way to do it but be aware of the limitations and consequences and please mention them in your manuscript.

-          Lind 209: What does WHT5R mean? It is not explained in the text.

-          Table 5: Are the exact p-values necessary or is it sufficient to use p<0.001? And please highlight significant p-values e.g. in bold.

-          Discussion: Add the before-mentioned limitations.

Best regards

Author Response

Thank you for your detailed review, I have attached a document containing your original comments and our response.

Reviewer 3 Report

In the future research, the number of participants could be increased. Including multi-ethnic and multi-regional nationalities.

Author Response

Thank you for your review, we agree with your comment. We have stated in the discussion that future studies should use appreciably larger sample sizes to fully assess predictive capabilities. This includes increased variability with regards to ethnic diversity.

Reviewer 4 Report

Overall, the paper focused on an important topic on trying to find better predictors for body fat estimation. However, there are a few issues:

1.      Authors did not justify why only used male subjects for this study. As both gender and ethnicities both played important roles in fat distributions. Only used male subjects raise major concerns on the model performances to predict body fat magnitude and distribution for general population.   

2.      Please double check the reference. For example, reference 5 was incomplete, the publication year was missing.

3.      Only two references were used in the discussion was not sufficient. More references are needed in the discussion to compare current study with previous studies or making statements.

Author Response

  1. Authors did not justify why only used male subjects for this study. As both gender and ethnicities both played important roles in fat distributions. Only used male subjects raise major concerns on the model performances to predict body fat magnitude and distribution for general population.
    Agreed, we have added a limitations section to the discussion section which mentions the need for more studies -- especially for the general population. We intend to expand our work in the future but this initial study has provided evidence that further research is warranted
  2. Please double check the reference. For example, reference 5 was incomplete, the publication year was missing.
    Thanks, reference corrected and further detail added in others after a review
  3. Only two references were used in the discussion was not sufficient. More references are needed in the discussion to compare current study with previous studies or making statements.
    Agreed that not enough references to previous work have been made. I have been through the discussion and added more qualifying statements and citations to previous work throughout the discussion